# Applying Machine Learning to Determine 25(OH)D Threshold Levels Using Data from the AMATERASU Vitamin D Supplementation Trial in Patients with Digestive Tract Cancer

**DOI:** 10.3390/nu14091689

**Published:** 2022-04-19

**Authors:** Katharina Otani, Kazuki Kanno, Taisuke Akutsu, Hironori Ohdaira, Yutaka Suzuki, Mitsuyoshi Urashima

**Affiliations:** 1Division of Molecular Epidemiology, The Jikei University School of Medicine, 3-25-8 Nishi-Shimbashi, Minato-ku, Tokyo 105-8461, Japan; katharina@jikei.ac.jp (K.O.); k.k.effort.patience@gmail.com (K.K.); taisuke0107.jusom@gmail.com (T.A.); 2Siemens Healthcare K.K., Osaki Gate City West Tower, 1-11-1 Osaki, Shinagawa-ku, Tokyo 114-8644, Japan; 3Department of Surgery, International University of Health and Welfare Hospital, 537-3 Iguchi, Nasushiobara 329-2763, Japan; ohdaira@iuhw.ac.jp (H.O.); yutaka@iuhw.ac.jp (Y.S.)

**Keywords:** machine learning, multivariate adaptive regression splines, vitamin D, 25(OH)D, parathyroid hormone, digestive tract cancer, RCT

## Abstract

Some controversy remains on thresholds for deficiency or sufficiency of serum 25-hydroxyvitamin D (25(OH)D) levels. Moreover, 25(OH)D levels sufficient for bone health might differ from those required for cancer survival. This study aimed to explore these 25(OH)D threshold levels by applying the machine learning method of multivariable adaptive regression splines (MARS) in post hoc analyses using data from the AMATERASU trial, which randomly assigned Japanese patients with digestive tract cancer to receive vitamin D or placebo supplementation. Using MARS, threshold 25(OH)D levels were estimated as 17 ng/mL for calcium and 29 ng/mL for parathyroid hormone (PTH). Vitamin D supplementation increased calcium levels in patients with baseline 25(OH)D levels ≤17 ng/mL, suggesting deficiency for bone health, but not in those >17 ng/mL. Vitamin D supplementation improved 5-year relapse-free survival (RFS) compared with placebo in patients with intermediate 25(OH)D levels (18–28 ng/mL): vitamin D, 84% vs. placebo, 71%; hazard ratio, 0.49; 95% confidence interval, 0.25–0.96; *p* = 0.04. In contrast, vitamin D supplementation did not improve 5-year RFS among patients with low (≤17 ng/mL) or with high (≥29 ng/mL) 25(OH)D levels. MARS might be a reliable method with the potential to eliminate guesswork in the estimation of threshold values of biomarkers.

## 1. Introduction

Vitamin D is a steroid hormone that has long been known to help the body absorb and retain calcium and phosphorus; both are critical for building bone. Thus, vitamin D deficiency in children and in adults causes rickets and osteoporosis, respectively, and increases the risk of fracture. On the other hand, cancer cells are believed to take up and activate 25-hydroxyvitamin D (25[OH]D) within the cell, which binds to the vitamin D receptor to regulate gene expression and consequently suppresses cancer growth. Thus, cancer patients with vitamin D deficiency have a short survival, conversely, vitamin D supplementation may be suggested to improve the prognosis of patients with cancer.

Definitions of normal levels of 25(OH)D remain debatable [1,2,3], because the reported normal levels of 25(OH)D appear to depend on the studied population [4]. Moreover, serum 25(OH)D levels sufficient for bone health [5,6,7] might differ from those required for preventing cancer death [8], and various beneficial effects of 25(OH)D in relation to cancer have been hypothesized, including a protective effect against cancer and a prolonging effect on relapse-free survival (RFS) [9,10]. 

For bone health, 25(OH)D levels that ensure high calcium absorption without causing toxicity while also maintaining low serum levels of parathyroid hormone (PTH) are desirable [11,12,13]. To reach these targets, thresholds to define vitamin D deficiency that range from 12 to 40 ng/mL 25(OH)D have been proposed [1,14], and the consensus is that serum levels below 12 ng/mL indicate deficiency, while levels above 30 ng/mL can most likely be considered sufficient [3]. However, the controversy remains regarding toxicity levels [3]. 

To increase the precision of 25(OH)D estimates in vitamin-D-related studies, efforts are being made to standardize assays [3,15]. Moreover, methodologies for evaluating physiological ranges of nutrients have also been recommended [16], but only a few investigators have evaluated the statistical methods applied to data analysis [17,18]. Machine learning applications have become available in many statistical software packages. The advantages of these applications are that they can be applied to a broad range of data distributions, require less a priori assumptions and offer more flexibility than conventional modeling [19].

We applied the machine learning method of multivariate adaptive regression splines (MARS) [20] to explore threshold serum 25(OH)D levels in patients who were surgically treated for digestive tract cancer and were enrolled in the AMATERASU randomized clinical trial on vitamin D [9]. The AMATERASU trial investigated the effects of vitamin D supplementation on relapse-free survival of patients with digestive cancer. Among the 417 patients with digestive tract cancers (from esophagus to rectum) who were enrolled, the 5-year relapse-free survival rate for those randomized to vitamin D, 2000 IU/d, vs. placebo was 77% vs. 69%, a difference that was not statistically significant [9]. Here, post hoc analyses were conducted on the impact of vitamin D supplementation on serum calcium and PTH levels in patients who were known to potentially suffer from osteoporosis or osteomalacia after surgery [21,22,23] and who were stratified according to the thresholds estimated by MARS, and on survival in subgroups of patients stratified according to the thresholds suggested by MARS.

## 2. Materials and Methods

### 2.1. Trial Design and Participants

This study was a post hoc analysis of data collected for the AMATERASU randomized controlled trial (UMIN000001977) of the effects of vitamin D_3_ supplementation on survival in patients with digestive tract cancer [9]. The ethics committees of the International University of Health and Welfare Hospital, Otawara, Tochigi, Japan, and the Jikei University School of Medicine, Nishi-Shimbashi, Tokyo, Japan, approved the trial (ethical approval codes 13-B-263 and 21-216 (6094), respectively). All enrolled patients provided written informed consent before their surgical treatment.

The double-blind AMATERASU trial ran from January 2010 to February 2018, investigating whether RFS was longer in patients who took vitamin D_3_ supplementation than those who took a placebo. Patients were randomized at their first outpatient visit to the hospital 2–4 weeks after surgical treatment. The allocation ratio was 3:2 to vitamin D_3_ supplementation (2000 IU/day) versus placebo groups. Both medications were manufactured by Zenyaku Pharmaceutical Co. Ltd. (Tokyo, Japan) to look and smell the same, to ensure blinding of patients and treating surgeons during the trial.

Follow-up examinations, including blood tests, were scheduled monthly for the first 6 months, then bimonthly for the next 6 months and every 3 months until the end of the trial. The blood tests included measurements of levels of calcium as appropriate and annual assessments of PTH. Vitamin D_3_ levels were quantified as serum 25(OH)D concentrations measured every year at approximately the same month by radioimmunoassay (SRL Inc., Hachioji, Tokyo, Japan) [24].

For estimating thresholds by applying MARS, we used data from all patients one year after starting either vitamin D or placebo supplementation, because serum 25(OH)D levels were expected to vary widely in the entire patient cohort due to the vitamin D supplementation. To validate the thresholds set by MARS, we used clinical data at both baseline and one year after randomization. Normal ranges of biomarkers were defined as 8.8–10.4 mg/dL for calcium and 10–65 pg/mL for intact PTH, according to the guidelines of the hospital that conducted the trial.

### 2.2. Statistical Analysis

Categorical data were compared by the Fisher’s exact test. Continuous data were compared by the paired or two sample Student’s *t*-test depending on whether data from the same patients or from two different groups of patients were used. The assumptions of normality were tested by bootstrapping the *t*-statistics, then inspecting the *t*-statistics distribution for normality on quantile–quantile plots and using the Shapiro–Wilk test as a quantitative test [25]. The assumptions of homogeneity of variance were tested by Levene’s test [26].

Associations between serum 25(OH)D levels and calcium and PTH levels were investigated by applying two machine learning methods and comparing the results with linear regression models. Residuals were inspected to confirm the fit of linear regression models.

First, we applied the machine learning method of locally weighted regression and scatterplot smoothing (LOWESS) for visual assessment of the associations of each biomarker, i.e., calcium and PTH with 25(OH)D [27]. LOWESS generates a smoothed line through a scatter plot that closely follows the data, thereby revealing the trends of data not easily seen in a scatter plot. Each point of the LOWESS plot is calculated through a weighted linear interpolation using neighboring points. The proportion of included points is fixed by selecting a bandwidth between 0 and 1. We used the default bandwidth value of 0.8, i.e., 80% of the data were included in the calculation of each point of the LOWESS plot. We qualitatively examined the LOWESS plots and visually assessed whether they could be approximated by segments of straight lines.

Next, we applied the machine learning method of multivariate adaptive regression splines (MARS) that fits regression models with hinge points and does not require specific data distributions nor any predefinition of the number or location of hinge points [20]. We used MARS to estimate limits (hinges) between segments of 25(OH)D plots where calcium or PTH slopes changed. The calculated models thus consisted of segments of differing slopes (coefficients) before and after hinge points. The procedure uses both a forward pass, which starts with a model with zero hinges, and a backward pass, which starts with an overfitted model with several hinges, and calculates the best fitting model based on least square regression. To avoid overfitting, we included a 10-fold validation procedure that randomly divided data into ten folds, using nine folds for fitting the model and the remaining fold for validation. This was repeated 100 times to obtain the final models for calcium versus 25(OH)D and PTH versus 25(OH)D.

Finally, we used piecewise linear regression analyses to fit straight lines through segments seen on the LOWESS plots to quantify the associations between 25(OH)D and calcium and PTH. In order to determine where to divide the data for regression analysis of segments, we fitted 25 linear regression lines for ranges of 25(OH)D from its minimum value up to a value between 15 and 40 ng/mL in steps of 1 ng/mL, using R-squared as the index variable. The remaining observations were used to fit a second linear regression segment of the near plateau region of higher 25(OH)D values. This method provided estimates for limits between segments, although the fitted lines did not necessarily meet, and generated plots that were not continuous. To obtain continuous plots, we calculated the intersection points of each of two segments.

Relapse- and death-related outcomes were assessed according to the randomization group, i.e., whether or not supplements were taken. Subgroups were stratified into either higher or lower than the threshold 25(OH)D level determined by MARS. The effects of vitamin D and placebo on the risk of relapse or death were estimated using RFS and Nelson–Aalen cumulative hazard curves for outcomes. A Cox proportional hazards model was used to determine the hazard ratio (HR) and 95% confidence interval (CI). To clarify whether vitamin D supplementation significantly affected these subgroups, a Cox regression model was applied that included three variables: (1) vitamin D group; (2) high 25(OH)D level group; and (3) vitamin D group and high 25(OH)D group multiplied together as an interaction variable. Two-way interaction tests were used to compare subgroups. Test results with two-sided *p* < 0.05 were considered significant.

R (RStudio, Version 1.4.1717, R package earth: Multivariate Adaptive Regression Splines, Version 5.3.1) was used for MARS and Stata (Stata 17, StataCorp LLC, College Station, TX, USA) was used for all other statistical analyses.

## 3. Results

### 3.1. Patient Characteristics

The AMATERASU trial enrolled 417 patients with digestive tract cancer, of whom 251 patients were given vitamin D supplementation and 166 were given placebo. The flow diagram showing patient participation in the trial and details of the patients’ characteristics at baseline have been published elsewhere [9]. Here, we analyzed biomarkers of all patients at year 1 of the study, as shown in the flow diagram (Figure 1). Baseline 25(OH)D values were missing for seven patients (four in the placebo group, three in the vitamin D group) and 25(OH)D values at one year were not measured for 60 patients (23 in the placebo group, 37 in the vitamin D group) due to death, transfer to distant hospitals, and other reasons.

The mean age of the patients analyzed one year after randomization was 66.3 ± 10.7 years at entry. None of the patients was on dialysis (mean ± SD creatinine value of 0.78 ± 0.18 mg/dL). Patient characteristics at year 1 of the study are shown in Table 1.

As expected, 25(OH)D levels were higher in the vitamin D group than in the placebo group. Calcium levels were the same between the two groups. On the other hand, PTH levels were lower in the vitamin D group than in the placebo group.

### 3.2. Threshold of Serum 25(OH)D Levels Relative to Calcium

The LOWESS plot (Figure 2A) suggested that serum calcium levels increased with increasing levels of serum 25(OH)D up to a threshold 25(OH)D level of approximately 20 ng/mL, after which calcium levels appeared to reach a plateau almost independent of 25(OH)D, then increased again from 25(OH)D levels of approximately 55 ng/mL.

As the best model, MARS calculated a fitted line with hinge 25(OH)D values of 17 ng/mL and 47 ng/mL (Figure 2B). The R-squared value of the model was 0.08. Up to the first hinge, the estimated calcium level increased by 0.04 mg/dL calcium for each 1 ng/mL increase in 25(OH)D. For 25(OH)D values between 17–47 ng/mL, estimated calcium levels stayed constant at 9.3 mg/dL. For 25(OH)D levels >47 ng/mL, estimated calcium levels increased again at the rate of 0.006 mg/dL calcium for each 1 ng/mL increase in 25(OH)D levels.

The linear regression analysis of calcium levels using the lower values of 25(OH)D had a maximum R-squared of 0.14 for a model including 25(OH)D values up to 23 ng/mL. Above this value, the mean calcium level increased by 0.035 mg/dL for each 1 ng/mL increase in 25(OH)D levels (95% CI 0.018–0.051 mg/dL, *p* < 0.001). The regression model using the remaining observations for 25(OH)D > 23 ng/mL revealed a near plateau segment with an estimated increase in calcium levels of close to zero (0.002 mg/dL for each 1 ng/mL increase in 25(OH)D, 95% CI −0.0006–0.0005 mg/dL, *p* = 0.12). The intersection point of the regression lines was at 17.4 ng/mL for 25(OH)D and 9.3 mg/dL (95% CI: 9.2–9.4 mg/dL) for calcium (Figure 2C).

We attempted to estimate a second intersection point around 50 ng/mL 25(OH)D as seen on the LOWESS and MARS plots by linear regression using the same methodology. The point above which the regression model had a maximum R-squared value (0.12) was found at 66 ng/mL for 25(OH)D, although the slope of this segment did not reach significance (*p* = 0.26), and the intersection point of this segment with the near plateau segment between 23–66 ng/mL was estimated at a clinically unrealistic value of −0.0012 ng/mL, indicating that piecewise linear regression might not discriminate a second threshold.

### 3.3. Threshold of 25(OH)D Relative to PTH

The LOWESS plot (Figure 3A) suggested that PTH levels decreased with increasing serum 25(OH)D levels up to the threshold 25(OH)D level, above which PTH levels appeared to decrease less steeply and nearly independent of 25(OH)D levels. PTH levels appeared to decrease again with 25(OH)D levels above 70 ng/mL.

As the best model, MARS calculated a fitted line with a hinge at the 25(OH)D value of 29 ng/mL (Figure 3B). The R-squared value of the model was 0.14. Up to this hinge, the estimated PTH levels decreased by 0.96 pg/mL per 1 ng/mL increase in 25(OH)D. For 25(OH)D values >29 ng/mL, the estimated PTH levels stayed constant at 20.1 pg/mL.

The linear regression for the lower range of 25(OH)D values reached a maximum R-squared of 0.14 for a threshold 25(OH)D value of 39 ng/mL (Figure 3C). The intersection point of the regression lines was 29.0 ng/mL for 25(OH)D and 42.7 pg/mL for PTH (95% CI: 40.06–45.4 pg/mL). The estimated negative slope of PTH for 25(OH)D levels >29.0 ng/mL did not reach significance (−0.09 pg/mL for each 1 ng/mL increase in 25(OH)D, 95% CI −0.29–0.10 pg/mL, *p* = 0.35), although it showed the same trend of decreasing PTH levels towards higher levels of 25(OH)D as revealed in the LOWESS plot.

### 3.4. Effect of Vitamin D Supplementation on Calcium Levels

Since the results of our analysis by MARS suggested threshold 25(OH)D levels of 17 ng/mL and 47 ng/mL, changes in calcium levels from baseline to one year after randomization were compared between placebo and vitamin D groups stratified according to these threshold levels (Figure 4). For patients with baseline 25(OH)D levels of ≤17 ng/mL, vitamin D supplementation increased estimated mean calcium levels by 0.20 mg/dL (95% CI 0.07–0.33 mg/dL, *p* = 0.002), whereas there was no significant alteration in calcium levels in the placebo group (Figure 4A). For patients with baseline 25(OH)D levels between 18–47 ng/mL, calcium levels remained at the same level in the vitamin D group and the placebo group (Figure 4B). There were insufficient numbers (*n* = 5) of patients with 25(OH)D levels of ≥48 ng/mL to analyze the impact of vitamin D supplementation on calcium.

### 3.5. Effect of Vitamin D Supplementation on PTH Levels

Since the results of our analysis by MARS suggested threshold 25(OH)D levels of 29 ng/mL, changes in PTH levels from baseline to one year after randomization were compared between placebo and the vitamin D groups stratified according to this threshold level (Figure 5). For patients with baseline 25(OH)D levels of either <29 ng/mL or ≥29 ng/mL, vitamin D supplementation did not significantly alter PTH levels.

### 3.6. Relapse-Free Survival in Subgroups Stratified according to Thresholds Determined by MARS

A total of 410 patients were stratified into two subgroups according to the threshold baseline 25(OH)D level of 17 ng/mL. The effects of vitamin D supplementation on RFS were compared between the high (>17 ng/mL) and low (≤17 ng/mL) 25(OH)D subgroups (Figure 6). Among patients with high 25(OH)D levels (*n* = 278), the 5-year RFS rate was 83% in the vitamin D group, which was significantly higher than the rate of 72% in the placebo group (HR, 0.56; 95% CI, 0.32–0.97) (Figure 6A). In contrast, among patients with low 25(OH)D levels (*n* = 132), RFS did not differ significantly between comparative groups (Figure 6B). There was a significant two-way interaction between the subgroups of high 25(OH)D levels and vitamin D supplementation (*p* for interaction = 0.03).

When the patients were stratified according to baseline 25(OH)D levels of 29 ng/mL (*n* = 77; ≥29 ng/mL), no significant associations were found in the same analyses (Figure 7).

The patients were divided into three subgroups stratified according to the threshold baseline 25(OH)D levels of 17 ng/mL and 29 ng/mL. The effects of vitamin D supplementation on RFS were compared with placebo in each of the subgroups of low (≤17 ng/mL), intermediate (18–28 ng/mL) and high (≥29 ng/mL) baseline 25(OH)D levels. As shown above, no significant effects of vitamin D supplementation were observed in the low (Figure 6B) and high 25(OH)D subgroups (Figure 7B). In contrast, among patients with intermediate 25(OH)D levels (*n* = 201), the 5-year RFS was 84% in the vitamin D group, which was significantly higher than 71% in the placebo group (HR, 0.49; 95% CI, 0.25–0.96; *p* = 0.04) (Figure 8). There was a significant two-way interaction between the subgroups of intermediate 25(OH)D levels and vitamin D supplementation (*p* for interaction = 0.03) compared with that in the low- and high-level groups.

## 4. Discussion

We estimated threshold serum levels of 25(OH)D required for maintenance of serum calcium and PTH levels by the machine learning method MARS and found two 25(OH)D threshold values for calcium (17 and 47 ng/mL) and one threshold value (29 ng/mL) for PTH. Our results supported the current consensus on threshold 25(OH)D levels required for maintenance of bone health [3], and also suggested that 25(OH)D thresholds might vary according to the biomarkers evaluated.

The question of whether PTH reaches a true plateau, however, remained unanswered. The threshold 25(OH)D value estimated by MARS corresponded with the results of standard methods of piecewise linear regression, although the results obtained by LOWESS and piecewise linear regression suggested that PTH might not reach a plateau value above a certain 25(OH)D threshold.

The results of our study did not provide a threshold for toxicity of vitamin D, although MARS calculated a second threshold point of 25(OH)D of 47 ng/mL for calcium. Toxicity has been confirmed for 25(OH)D levels above 110 ng/mL [13], which is outside the range seen in the patients enrolled in this trial. The threshold for toxicity may however be lower, as recent studies have suggested that toxicity in terms of increasing risk of falls might already exist with 25(OH)D values above 40–45 ng/mL [28,29].

A post hoc analysis of the AMATERASU trial on the impact of vitamin D supplementation on calcium showed that vitamin D supplementation does increase calcium levels in patients with low baseline 25(OH)D levels (≤17 ng/mL). This confirms previous observations that patients who have undergone surgery of the gastrointestinal tract, especially gastrectomy, might benefit from vitamin D supplementation, since they may besusceptible to developing osteoporosis and osteomalacia [21,22,23,30,31]. The impact of vitamin D supplementation on PTH levels in the present study were inconclusive, although these results were likely to have been due to the confounding effects of other factors. First, the baseline levels of biomarkers were measured on the first day of the patients’ outpatient visits to the hospital, 2 to 4 weeks after operation, and follow-up measurements were performed one year later. During this one-year time period, patients naturally recovered, which could have resulted in changes in biomarker levels regardless of vitamin D supplementation. Second, since most patients enrolled in the trial had calcium and PTH levels in the normal range, the effect of vitamin D supplementation on these biomarkers was expected to remain small.

Systematic literature reviews on threshold values of 25(OH)D that suppress high PTH levels have been published by Aloia et al. in 2006 and Sai et al. in 2011 [32,33]. Further investigations on whether PTH reaches near plateau values in various populations have been performed since then [34,35,36,37]. The proposed 25(OH)D threshold values ranged from 10–49 ng/mL [38,39], although some authors observed no threshold value above which PTH reaches a plateau [40]. It is, thus, challenging to define a threshold in a gradually changing curve. Furthermore, since the normal range of PTH is wide (10–65 pg/mL for intact PTH), one could argue about where the near plateau commences.

Several methods have been applied to define 25(OH)D thresholds. Earlier studies visually determined 25(OH)D thresholds based on stratified data [5,41,42], although they were associated with loss of information due to categorization; additionally, visual assessments might be investigator dependent. Exponential decay curves and quadratics have also been used to explain the relationship between 25(OH)D and PTH [32,43,44]. These models can be applied to continuous data and make better use of all the information available in the data. Piecewise regression models that fit two or more connected linear segments with differing slopes to estimate knot points or thresholds have also been reported in several studies [12,17,32,45,46,47].

With the advancement of computer science, new data-driven methods of data analysis have become available. Locally weighted regression smoothing (LOESS or LOWESS) lines might reveal trends of data that might not be otherwise easily discovered, to guide the selection of statistical models subsequently used to quantify the relationship between 25(OH)D and PTH [17,32,48]. Machine learning methods are becoming popular to analyze medical data. A regression tree approach has been applied to investigate the relationship between 25(OH)D and PTH [18], although the results might vary depending on the stopping rules for tree growth. In this study, we applied MARS to overcome some of the challenges described above, since MARS offers the advantage of eliminating a priori assumptions, such as the number and location of thresholds.

We previously showed that vitamin D supplementation improved RFS in the prespecified subgroup of patients with intermediate (20–40 ng/mL) baseline serum 25(OH)D levels, but not in those with low (<20 ng/mL) or high (>40 ng/mL) serum 25(OH)D levels [9]. In this post hoc analysis using the same data, vitamin D supplementation significantly improved 5-year RFS in the subgroup of patients with intermediate (18–28 ng/mL) 25(OH)D levels, which seems to represent more precise estimations than the prespecified cutoff points [9]. In our previous analysis [9], since only five patients had high levels of 25(OH)D at baseline, the number was too few to allow statistical evaluation. In contrast, this study clarified that vitamin D supplementation was not effective in 77 patients with high 25(OH)D levels (≥29 ng/mL). This suggests that serum 25(OH)D levels of approximately 30 ng/mL and above can be considered to be sufficient, and that vitamin D supplementation for prevention of cancer relapse might not be needed in patients with these serum 25(OH)D levels. On the other hand, it was hypothesized that vitamin D supplementation would be effective in the subgroup with low 25(OH)D levels at baseline, although it was not effective in terms of preventing relapse in patients with low baseline 25(OH)D levels (≥17 ng/mL). It is possible that the supplementary dosage of 2000 IU/d used in the trial might have been insufficient to increase 25(OH)D levels sufficiently in these patients. This suggests that deficient or sufficient levels of 25(OH)D for bone health [5,6,7] might differ from the levels required for preventing cancer relapse or death [8].

This study has several limitations. First, patients underwent blood sampling to measure serum total 25(OH)D levels at the first outpatient visit between 2 and 4 weeks after operation, but not preoperatively. Serum levels of 25(OH)D have been reported to decrease after surgery [49,50]. Thus, baseline levels of 25(OH)D could have been affected by the stress induced by surgery and cannot be exactly compared with 25(OH)D levels in other studies in which blood sampling took place before surgery or before a diagnosis of cancer. Second, MARS was only applied for serum 25(OH)D levels and not for bioavailable 25(OH)D levels, the measurement in cancer patients of which is reasonable [51]. Third, only five patients had 25(OH)D levels higher than 47 ng/mL, which was too few to allow a statistical evaluation for intoxication. Fourth, this study performed an exploratory analysis that was not prespecified in the original protocol of the AMATERASU trial and must, therefore, be interpreted with caution. Fifth, because all the patients included in this study were Japanese, most esophageal cancers were squamous cell carcinomas, the incidence of gastric cancer was still relatively high, and the optimal levels of 25(OH)D could be different from those in other population groups. Thus, the results of the present study are not necessarily generalizable to other populations. Sixth, the study population included patients with a mixture of cancers with biological and clinical differences. This study also had specific strengths. First, through the AMATERASU trial, data from a homogenous group of patients could be collected who presented with a wide range of 25(OH)D levels. Second, the thresholds obtained by applying machine learning could be tested as a post hoc analysis of a randomized trial where the two groups of patients differed only by receiving vitamin D supplementation or placebo.

## 5. Conclusions

In conclusion, in patients who have undergone surgery for digestive cancer, vitamin D supplementation might be recommended for bone health in those with low baseline 25(OH)D levels of ≤17 ng/mL, and for improving cancer survival in those with intermediate baseline 25(OH)D levels of 17–29 ng/mL, although it should not be given to patients with baseline 25(OH)D levels close to 47 ng/mL to avoid potential toxicity. MARS might be a promising analytical method for a more precise estimation of thresholds for specific outcomes.

## Figures and Tables

**Figure 1 nutrients-14-01689-f001:**
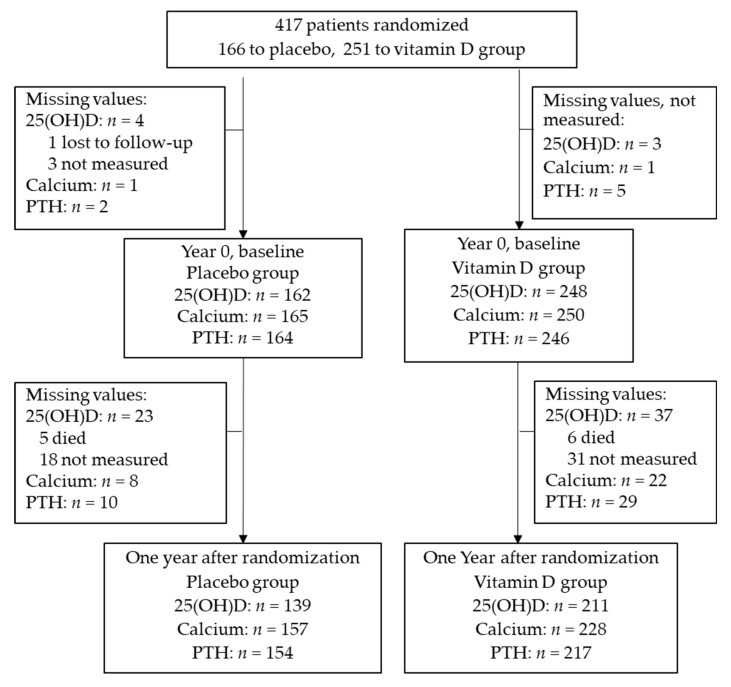
Flow diagram of patient enrolment (25(OH)D, 25-hydroxyvitamin D; PTH, parathyroid hormone).

**Figure 2 nutrients-14-01689-f002:**
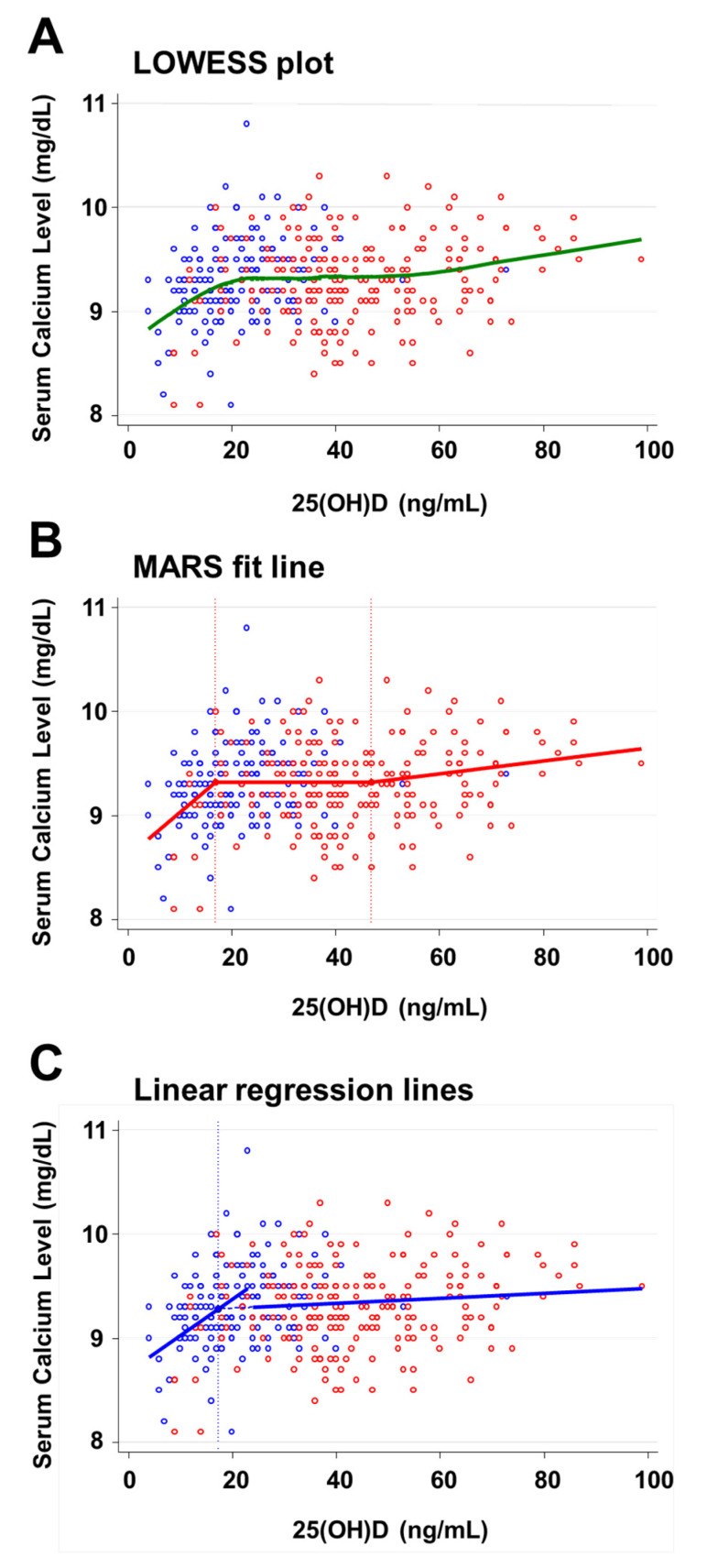
Calcium levels relative to 25(OH)D levels in all patients who received a placebo or vitamin D (**A**) overlaid with the LOWESS plot, (**B**) overlaid with the MARS fitted line with hinge points at 17 and 47 ng/mL for 25(OH)D and a plateau value of 9.3 mg/dL for calcium, and (**C**) overlaid with linear regression lines with points of intersection at 17.3 ng/mL for 25(OH)D and 9.3 mg/dL for calcium (95% CI 9.2–9.4 mg/dL), and a limit between fitted segments of 23 ng/mL for 25(OH)D. Serum calcium levels from the placebo group and vitamin D group are shown as blue and red circles, respectively.

**Figure 3 nutrients-14-01689-f003:**
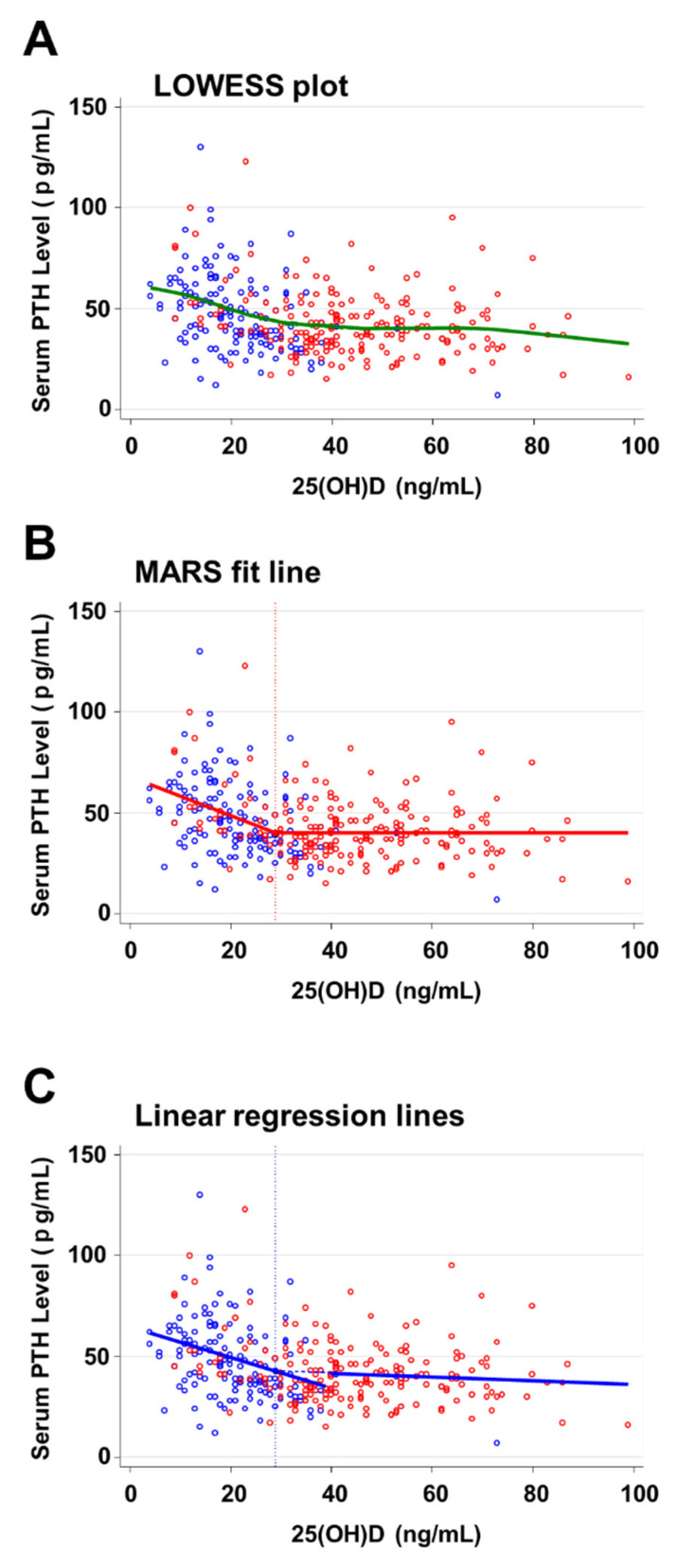
PTH levels relative to 25(OH)D levels in all patients receiving placebo or vitamin D (**A**) overlaid with the LOWESS plot, (**B**) overlaid with the MARS fitted line with a hinge point at 39 ng/mL, and (**C**) overlaid with linear regression lines with the point of intersection for 25(OH)D of 29.0 ng/mL and limit between fitted segments of 39 ng/mL for 25(OH)D. Serum PTH levels from the placebo group and vitamin D group are shown as blue and red circles, respectively.

**Figure 4 nutrients-14-01689-f004:**
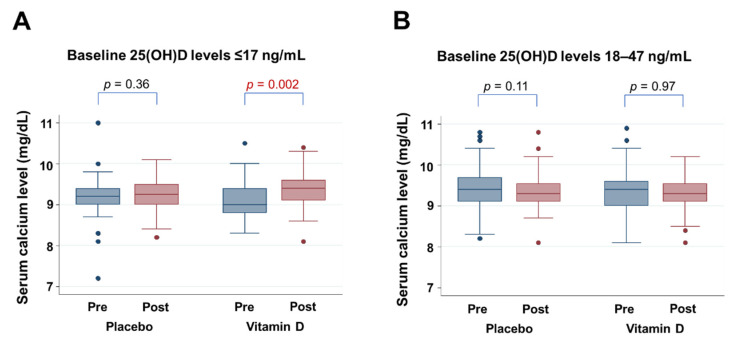
Calcium levels at baseline and at year 1 into the trial, in groups with placebo and vitamin D supplementation (**A**) for patients with baseline 25(OH)D levels of 17 ng/mL and less, (**B**) for patients with baseline 25(OH)D levels of between 18 and 47 ng/mL. One case with low calcium levels (5.9 mg/dL) in the vitamin D group before supplementation is not presented, since it was an outlier.

**Figure 5 nutrients-14-01689-f005:**
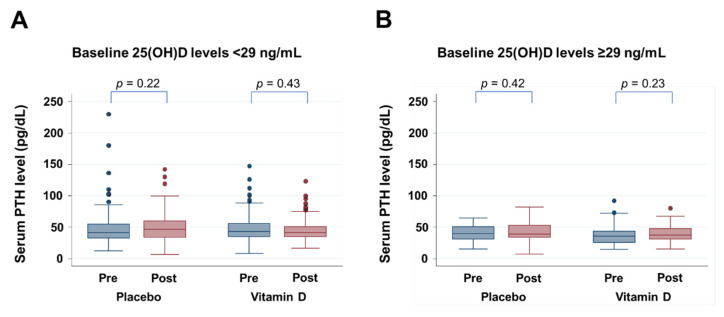
PTH levels at baseline and at year 1 into the trial with placebo and vitamin D supplementation (**A**) in patients with baseline 25(OH)D levels of <29 ng/mL, and (**B**) patients with baseline 25(OH)D levels ≥29 ng/mL.

**Figure 6 nutrients-14-01689-f006:**
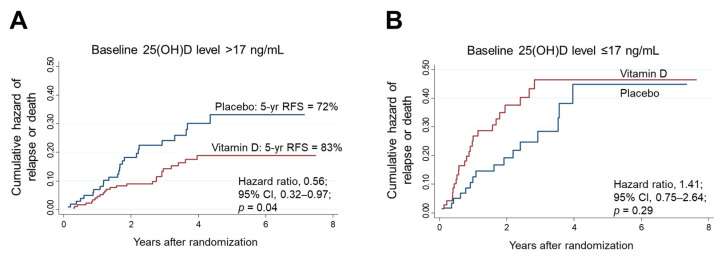
Nelson–Aalen cumulative hazard curves for relapse or death in the subgroups of 25(OH)D levels higher (**A**) and lower (**B**) than the threshold level (17 ng/mL) determined by MARS. 25(OH)D: 25-hydroxyvitamin D; CI: confidence interval; MARS: multivariate adaptive regression splines.

**Figure 7 nutrients-14-01689-f007:**
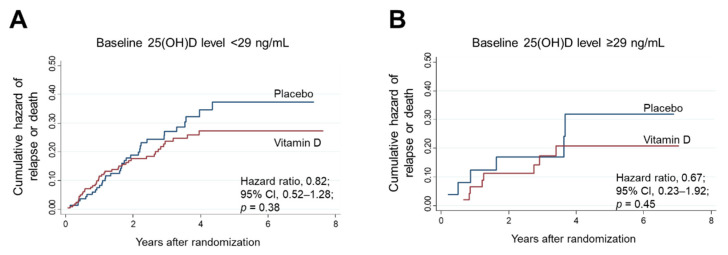
Nelson–Aalen cumulative hazard curves for relapse or death in the subgroups of 25(OH)D levels lower (**A**) and higher (**B**) than the threshold baseline level (29 ng/mL) determined by MARS. 25(OH)D: 25-hydroxyvitamin D; CI: confidence interval; MARS: multivariate adaptive regression splines.

**Figure 8 nutrients-14-01689-f008:**
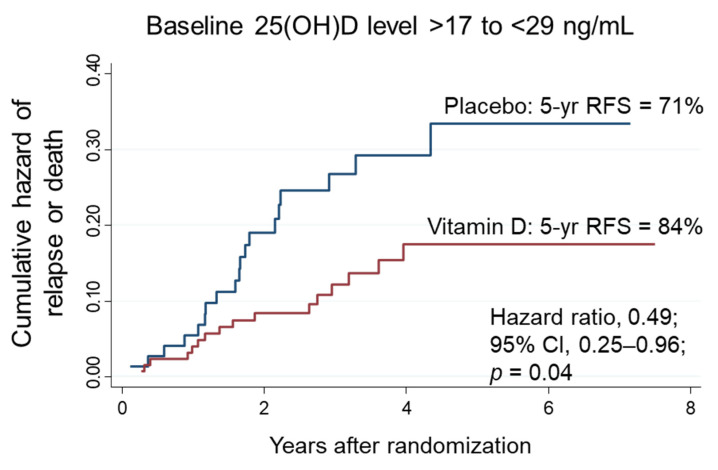
Nelson–Aalen cumulative hazard curves for death in the subgroups of intermediate 25(OH)D levels (higher than 17 ng/mL and lower than 29 ng/mL). The thresholds were determined by MARS for sufficiency of serum calcium and PTH levels, respectively. yr: years; 25(OH)D: 25-hydroxyvitamin D; CI: confidence interval.

**Table 1 nutrients-14-01689-t001:** Patient characteristics at one year after randomization (mean value ± standard deviation or number (%)).

Biomarker	Total	Placebo Group	Vitamin D Group	*p*-Value (between Groups)
Age at entry (year)	66.3 ± 10.7	64.2 ± 10.4	67.6 ± 10.5	0.001
Sex				0.07
Male (%)	233 (65.3)	85 (59.4)	148 (69.2)	
Female (%)	124 (34.7)	58 (40.6)	66 (30.8)	
BMI (kg/m^2^)	22.0 ± 3.5	21.9 ± 3.2	22.1 ± 3.7	
Cancer site				0.98
Esophagus (%)	33 (9.2)	14 (9.8)	19 (8.9)	
Stomach (%)	152 (42.6)	61 (42.7)	91 (42.5)	
Small intestine (%)	2 (0.6)	1 (0.7)	1 (0.5)	
Colorectum (%)	170 (47.6)	67 (46.9)	103 (48.1)	
Stage				0.36
I (%)	162 (45.4)	59 (41.3%)	103 (45.4)	
II (%)	96 (26.9)	43 (30.1)	53 (24.8)	
III (%)	99 (27.7)	41 (28.7)	58 (27.1)	
25(OH)D ^a^ (ng/mL)	34.9 ± 18.5	21.3 ± 9.7	44.0 ± 17.3	<0.001
Ca ^b^ (mg/dL)	9.3 ± 0.4	9.3 ± 0.4	9.3 ± 0.4	0.68
PTH ^c^ (pg/mL)	44.8 ± 17.8	47.4 ± 20.8	43.0 ± 16.5	0.03

^a^ 25-hydroxyvitamin D, ^b^ Calcium, ^c^ Parathyroid hormone.

## Data Availability

Data available on request from the authors.

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
