# Peer review of "Applying Machine Learning to Determine 25(OH)D Threshold Levels Using Data from the AMATERASU Vitamin D Supplementation Trial in Patients with Digestive Tract Cancer"

_nutrients, 2022, doi:10.3390/nu14091689_

Round 1

Reviewer 1 Report

First of all, I would like to congratulate the authors of a very interesting work using new objective methods of assessing vitamin D concentration and making an attempt to determine the correct vitamin D thresholds in various clinical contexts.

I have only two comments on the manuscript:

First of all, the study lacked information about the doses of vitamin D received by the patients.

Secondly, I cannot agree with the authors' opinion contained in the following lines: 321-323. In my opinion, the authors did not prove the toxicity of vitamin D above these values.

Author Response

First of all, I would like to congratulate the authors of a very interesting work using new objective methods of assessing vitamin D concentration and making an attempt to determine the correct vitamin D thresholds in various clinical contexts.

Thank you very much for the positive comment and suggestions.

I have only two comments on the manuscript:

First of all, the study lacked information about the doses of vitamin D received by the patients.

We agree that this is an important piece of information. We reported the value (2,000 IU/day) in a sentence in the Materials and Methods section.

Page 2, line 88

“The allocation ratio was 3:2 to vitamin D3 supplementation (2000 IU/day) versus placebo groups.”

Secondly, I cannot agree with the authors' opinion contained in the following lines: 321-323. In my opinion, the authors did not prove the toxicity of vitamin D above these values.

Thank you for this thoughtful comment. We agree that our sentences were overstating the result of a potential toxicity level of vitamin D and rephrased the paragraph.

Page 12, line 329

“The results of our study did not provide a threshold for toxicity of vitamin D although MARS calculated a second threshold point of 25(OH)D of 47 ng/mL for calcium. Toxicity has been confirmed for 25(OH)D levels above 110 ng/mL [13], which is outside the range seen in the patients enrolled in this trial. The threshold for toxicity may however be lower, as recent studies suggested that toxicity in terms of increasing risk of falls might already exist with 25(OH)D values above 40–45 ng/mL [28,29].”

Reviewer 2 Report

This study aimed to explore 25(OH)D threshold levels by applying the machine learning method of multivariable adaptive regression splines (MARS) in post-hoc analyzes using data from the AMATERASU trial, with digestive tract cancer to receive vitamin D or placebo supplementation.

In the introduction you could briefly include the results of the AMATERASU trial

It is not clear to me why it only includes patients with digestive cancer and not all patients in the study.

In figure 1, the patients who remain in each phase should appear in the central line and the patients who are not included with their causes should appear on the lateral line. Those two boxes with the remaining patients in each of the lines (treatment and placebo) are missing.

In table 1 there are only the data of the general group, there should also be a column with the treatment group and another column with the placebo group and its comparison of means (as in table 2). I think that tables 1 and 2 should be joined

In figures 2 and 3, the total group, the placebo group and the treatment group should be differentiated in each of the graphs.

In the discussion, you could compare the results obtained in this subgroup of patients against the overall study.

The discussion could end with the strengths of the study. It is very sad to end with so many weak points of the study

Author Response

This study aimed to explore 25(OH)D threshold levels by applying the machine learning method of multivariable adaptive regression splines (MARS) in post-hoc analyzes using data from the AMATERASU trial, with digestive tract cancer to receive vitamin D or placebo supplementation.

In the introduction you could briefly include the results of the AMATERASU trial

We agree that this is important information and included a sentence summarizing the results of the AMATERASU trial in the introduction.

Page 2, line 65

“The AMATERASU trial investigated the effects of vitamin D supplementation on relapse-free survival of patients with digestive cancer. Among the 417 patients with digestive tract cancers (from esophagus to rectum) who were enrolled, the 5-year relapse-free survival rate for those randomized to vitamin D, 2000 IU/d, vs placebo was 77% vs 69%, a difference that was not statistically significant [9].”

It is not clear to me why it only includes patients with digestive cancer and not all patients in the study.

We are sorry for this misunderstanding. The AMATERASU trial included only patients with digestive cancer. We added a sentence to clarify that only patients with digestive cancer were included in the AMATERASU trial.

Page 2, line 65

“The AMATERASU trial investigated the effects of vitamin D supplementation on relapse-free survival of patients with digestive cancer.”

In figure 1, the patients who remain in each phase should appear in the central line and the patients who are not included with their causes should appear on the lateral line. Those two boxes with the remaining patients in each of the lines (treatment and placebo) are missing.

Thank you for the suggestion. We modified figure 1 accordingly.

Page 4, line 174

In table 1 there are only the data of the general group, there should also be a column with the treatment group and another column with the placebo group and its comparison of means (as in table 2). I think that tables 1 and 2 should be joined

Thank you for the suggestion. We added columns for the placebo and vitamin D groups and joined table 1 and 2.

Page 5, line 178

Table 1. Patient characteristics at one year after randomization (mean value ± standard deviation or number (%))

Biomarker

Total

Placebo group

Vitamin D group

P-value

(between groups)

Age at entry (y)

66.3 ± 10.7

64.2 ± 10.4

67.6 ± 10.5

0.001

Sex

0.07

Male (%)

233 (65.3)

85 (59.4)

148 (69.2)

Female (%)

124 (34.7)

58 (40.6)

66 (30.8)

BMI (kg/m2)

22.0 ± 3.5

21.9 ± 3.2

22.1 ± 3.7

Cancer site

0.98

Esophagus (%)

33 (9.2)

14 (9.8)

19 (8.9)

Stomach (%)

152 (42.6)

61 (42.7)

91 (42.5)

Small intestine (%)

2 (0.6)

1 (0.7)

1 (0.5)

Colorectum (%)

170 (47.6)

67 (46.9)

103 (48.1)

Stage

0.36

  I (%)

162 (45.4)

59 (41.3%)

103 (45.4)

  II (%)

96 (26.9)

43 (30.1)

53 (24.8)

  III (%)

99 (27.7)

41 (28.7)

58 (27.1)

25(OH)Da (ng/mL)

34.9 ± 18.5

21.3 ± 9.7

44.0 ± 17.3

<0.001

Cab (mg/dL)

9.3 ± 0.4

9.3 ± 0.4

9.3 ± 0.4

0.68

PTHc (pg/mL)

44.8 ± 17.8

47.4 ± 20.8

43.0 ± 16.5

0.03

In figures 2 and 3, the total group, the placebo group and the treatment group should be differentiated in each of the graphs.

Thank you for pointing out that the groups could be differentiated. We modified figure 2 and 3 and used blue markers for the placebo and red markers for the vitamin D group.

In the discussion, you could compare the results obtained in this subgroup of patients against the overall study.

Thank you for this comment. In this study, we aimed at obtaining threshold values for 25(OH)D relating to calcium and PTH in patients who underwent surgery for digestive cancer. For this purpose, it was important to have patients with a wide range of 25(OH)D levels regardless of the reason for the 25(OH)D levels. The ranges of 25(OH)D levels of each group separately would not be wide enough for a robust determination of thresholds. We believe that the new figures clarify this point.

The discussion could end with the strengths of the study. It is very sad to end with so many weak points of the study

We agree that it is sad that our study has many weak points. We followed the format of other manuscripts and placed the limitations at the end of the discussion. To end the discussion on in a more positive tone, we added two specific strengths of this study.

Page 13, line 408

“This study also had specific strengths. First, through the AMATERASU trial, data from a homogenous group of patients could be collected who presented with a wide range of 25(OH)D levels. Second the thresholds obtained by applying machine learning could be tested as post-hoc analysis of a randomized trial where the two groups of patients differed only by receiving vitamin D supplementation or placebo.”